# Impact and benefit-cost ratio of a program for the management of latent tuberculosis infection among refugees in a region of Canada

**Jacques Pépin**[1]*, **France Desjardins**[2], **Alex Carignan**[1], **Michel Lambert**[2], **Isabelle Vaillancourt**[2], **Christiane Labrie**[2], **Dominique Mercier**[2], **Rachel Bourque**[2], **Louiselle LeBlanc**[1]

**1** Department of Microbiology and Infectious Diseases, Université de Sherbrooke, Sherbrooke, Québec, Canada, **2** Clinique des Réfugiés, Centre Local de Services Communautaires, Sherbrooke, Québec, Canada

* jacques.pepin@usherbrooke.ca

## Abstract

### Introduction

The identification and treatment of latent tuberculosis infection (LTBI) among immigrants from high-incidence regions who move to low-incidence countries is generally considered an ineffective strategy because only $\approx$14% of them comply with the multiple steps of the 'cascade of care' and complete treatment. In the Estrie region of Canada, a refugee clinic was opened in 2009. One of its goals is LTBI management.

### Methods

Key components of this intervention included: close collaboration with community organizations, integration within a comprehensive package of medical care for the whole family, timely delivery following arrival, shorter treatment through preferential use of rifampin, and risk-based selection of patients to be treated. Between 2009–2020, 5131 refugees were evaluated. To determine the efficacy and benefit-cost ratio of this intervention, records of refugees seen in 2010–14 (n = 1906) and 2018–19 (n = 1638) were reviewed. Cases of tuberculosis (TB) among our foreign-born population occurring before (1997–2008) and after (2009–2020) setting up the clinic were identified. All costs associated with TB or LTBI were measured.

### Results

Out of 441 patients offered LTBI treatment, 374 (85%) were compliant. Adding other losses, overall compliance was 69%. To prevent one case of TB, 95.1 individuals had to be screened and 11.9 treated, at a cost of $16,056. After discounting, each case of TB averted represented $32,631, for a benefit-cost ratio of 2.03. Among nationals of the 20 countries where refugees came from, incidence of TB decreased from 68.2 (1997–2008) to 26.3 per

**Data Availability Statement:** All relevant data files are available from the Dryad database (doi.org/10.5061/dryad.n8pk0p2wm).

**Funding:** The authors received no specific funding for this work.

**Competing interests:** The authors have declared that no competing interests exist.

100,000 person-years (2009–2020). Incidence among foreign-born persons from all other countries not targeted by the intervention did not change.

## Conclusions

Among refugees settling in our region, 69% completed the LTBI cascade of care, leading to a 61% reduction in TB incidence. This intervention was cost-beneficial. Current defeatism towards LTBI management among immigrants and refugees is misguided. Compliance can be enhanced through simple measures.

## Introduction

Massive immigration from high-incidence countries has irreversibly altered the epidemiology of tuberculosis (TB) in Canada where >70% of cases now occur among foreign-born patients [1]. In 2019, 341,000 immigrants settled in Canada including 48,500 refugees [2]. Failure to address this specific population explains why, over the last 15 years, the overall incidence of TB and incidence in foreign-born individuals have stagnated around 5 and 15 per 100,000 per person-years, respectively, while decreasing below 1 per 100,000 in the Canadian-born non-indigenous [1]. The reservoir of immigrants with latent tuberculosis infection (LTBI) is huge and constantly expanding.

Lowering the incidence of TB in the home countries of future immigrants would be the ideal solution, but this may require a few decades. The current approach of Immigration, Refugees and Citizenship Canada (IRCC) is to require a chest X-ray (CXR) for all applicants for permanent residency aged ≥11 years and for temporary residents from targeted countries. Persons with radiological anomalies need additional examinations pre-landing and are under surveillance post-landing [3]. Perhaps because it is considered ineffective by many [4–8], there is no IRCC-mandated top-down system to identify and manage immigrants with LTBI, which is left to the discretion of local health institutions.

In the province of Quebec, the flow of immigrants is concentrated in the Montreal area, but efforts are made to 'regionalize' immigration, especially for refugees whose destination is determined by the government, at least for their first year in Canada. Here we will assess the performance, impact and benefit-cost ratio of a program for the diagnosis and treatment of LTBI among a subset of immigrants, those who arrived with a refugee status obtained prior to landing or were asylum seekers, and we will describe the evolution of TB among immigrants in the Estrie region (2020 population: 334,000, half of them in the city of Sherbrooke) east of Montreal, where a single institution (Centre Hospitalier Universitaire de Sherbrooke [CHUS]) provides all care for LTBI and TB. Our experience illustrates what can be achieved in smaller urban areas.

## Methods

### Foreign-born population and tuberculosis in Estrie

Statistics Canada provided data concerning the country of birth of all individuals living in Estrie from censuses or surveys performed in 1996, 2001, 2006, 2011 and 2016. Foreign-born individuals include immigrants, non-permanent residents (students, qualified workers) and respondents born abroad of Canadian parents. Intercensal estimates were calculated according to the rate of change between censuses for specific subpopulations. IRCC provided data on the

number and categories of immigrants, for each country, who settled in Estrie from 2009 to 2020.

Cases of TB, identified through the CHUS records, were defined as patients having *Mycobacterium tuberculosis* (or, in one case, *M. africanum*) detected by culture and/or polymerase chain reaction. Patients (mostly children) with presumed (unproven) tuberculosis were excluded.

## The refugee clinic

The Sherbrooke CDR opened in February 2009 within a primary care facility, staffed by two full-time nurses and part-time family physicians. It offers a comprehensive evaluation which aims to detect and manage not only LTBI but a range of infectious and non-infectious diseases. Fewer than 10% of patients are asylum seekers. The bulk of the patient load are refugees pre-selected by IRCC prior to landing, who are referred by two community organizations that provide help to incoming refugees: one for privately sponsored Afghans and a second for all others, who are mostly government sponsored. The latter organization provides translators.

All refugees undergo a tuberculin skin test (TST); those with a reaction ≥10 mm undergo a CXR. With other information, this enables a calculation of the lifetime risk for reactivation of LTBI [9, 10]. Adults with a risk ≥5% are referred to an infectious disease clinic, as are children with a TST ≥10 mm. For adults, the standard treatment for LTBI has been 4 months of rifampin during the entire study period while a course of 9 months of isoniazide (INH) was given to patients unable to receive rifampin because of intolerance or contraindication [11]. For children, the first-line drug was INH initially, and rifampin starting in 2018 [12]. The Quantiferon interferon-gamma release assay (IGRA) was used on an *ad hoc* basis, generally for non-African non-Afghan younger adults with a TST of 10–14 mm, and for children.

Adults started on rifampin were followed with an alanine aminotransferase at 4 and 8 weeks, the latter with a medical visit as well. Patients experiencing adverse reactions were instructed to contact their translator to arrange a medical visit. Instructions were provided in writing at the initial medical visit, in the refugee's own language, in addition to explanations given through the translator. Community pharmacists were instructed to provide rifampin for 1–2 months, to be renewed for a total of 4 months. For patients who complied with blood tests at 4 and 8 weeks, came to the medical visit at 8 weeks and tolerated the medication, no appointment was given at the end of treatment, until 2019 when such a medical visit was scheduled. Patients who were intolerant of rifampin were switched to INH for 9 months, with medical visits at 2, 4 and 6 months.

CDR records were reviewed to extract data for all refugees seen during 2010–14 and 2018–2019: demographics, TST and CXR results, estimation of the risk of reactivation, and referrals. Data were collected using an Excel spreadsheet. For referrals, CHUS records were reviewed to gather data on LTBI management: decision to treat or not, choice of treatment, change in treatment if any, and compliance. Patients were considered to have completed treatment if they were present at the last scheduled medical visit and/or if examination of pharmacy data confirmed that they had procured all their meds within the expected time frame.

## Benefit-cost analyses

Cost estimates (in 2020 Canadian dollars), from the perspective of the healthcare system, used data from the *Régie de l'assurance-maladie du Québec* for medical acts and drugs, the refugee clinic (*Clinique des réfugiés*, CDR) and the Estrie public health division for nurses and physicians time, and the Quebec ministry of health for hospital costs. Records of foreign-born individuals diagnosed with TB between 1997 and 2020 were reviewed to measure duration of

hospital stay and other data. Number of contacts with or without LTBI were gathered from public health files. Using such figures, we estimated the average cost of a case of active tuberculosis, as well as the cost of LTBI diagnosis and treatment.

To calculate the value over a 30-year period of cases of TB averted from the perspective of LTBI cases managed in 2020, we used the standard 1.5% discounting rate currently recommended in Canada, as well as a higher rate of 3% as a sensitivity analysis [13]. No discounting was used for cost of LTBI management, based on 2020 data. To estimate the distribution in time of cases potentially averted over this 30-year period, we assumed that without the intervention the incidence of TB would have been 128 per 100,000 in first year post-immigration, 37 per 100,000 in years 2–5, 17 per 100,000 thereafter with a very slow decay and used these rates as weights for each year from 2020 till 2049 [5, 14]. This distribution of cases over time was then used to calculate costs taking into consideration the discounting.

### Ethics statement

The study was approved by the 'Comité d'Éthique de la Recherche (CER) du Centre Intégré Universitaire de Santé et de Services Sociaux-Centre Hospitalier Universitaire de Sherbrooke', Project #2021–4184. As the study consisted of a review of medical records, the CER waived the need for individual informed consent. All methods were carried out in accordance with relevant guidelines and regulations.

## Results

### The foreign-born population and tuberculosis in Estrie

In 2016, 20,840 residents of Estrie had been born outside Canada: 17,575 immigrants, 2245 non-permanent residents, 1020 born abroad of Canadian parents. The number of foreign-born individuals more than doubled between 1996 and 2016 (Table 1). Meanwhile the Canada-born population increased by only 10%. There was a spectacular growth of immigrant populations originating from countries with moderate or high incidence of TB. Their profile also changed: among those who settled in 2009–2020 (Table 2), 50% were refugees (compared to 20–30% previously), 36% were economic immigrants (many from France), while 14% were sponsored by a relative.

### Incidence of tuberculosis

We compared the 12 years that preceded the CDR (1997–2008) to the 12 years that followed (2009–2020). For Canada-born patients, in 1997–2008 there were 30 cases of TB (age: 41–88 years, median 73) compared to 12 in 2009–2020 (age: 54–91 years, median 76). Among the

**Table 1. Evolution of foreign-born population (immigrants, non-permanent residents, and Canadians by birth), Estrie region, 1996–2016.**

| Region of birth | Population | | | | | |
| --- | --- | --- | --- | --- | --- | --- |
| | 1996 | 2001 | 2006 | 2011 | 2016 | Growth 2016 vs 1996 |
| Canada-born | 263,240 | 268,745 | 279,230 | 286,395 | 289,175 | +10% |
| All foreign-born | 10,060 | 10,775 | 14,725 | 16,410 | 20,840 | +107% |
| Americas and Caribbean (USA excluded) | 1360 | 1530 | 2620 | 3275 | 3470 | +155% |
| Africa | 775 | 905 | 2205 | 2860 | 4120 | +432% |
| Asia and the Middle East | 1085 | 1330 | 1915 | 2305 | 3365 | +210% |
| USA, Europe, or Oceania | 6840 | 7000 | 7960 | 7970 | 9885 | +45% |
| Proportion of foreign-born from low- or middle-income countries | 32% | 35% | 46% | 51% | 53% | |

**Table 2. Distribution of immigrants who settled in Estrie region between 2009 and 2020 according to incidence of tuberculosis in their home country and categories of immigrants [15].**

| | Incidence in their home country | | | | |
|---|---|---|---|---|---|
| | ≥100 per 100,000 | 50–99 per 100,000 | 30–49 per 100,000 | <30 per 100,000 | Total |
| Refugees[a] | 4430 (36.4%) | 115 (0.9%) | 855 (7.0%) | 660 (5.4%) | 6060 (49.9%) |
| Economic migrants | 905 (7.4%) | 950 (7.8%) | 730 (6.0%) | 1750 (14.4%) | 4335 (35.7%) |
| Familial reunifications | 475 (3.9%) | 340 (2.8%) | 270 (2.2%) | 675 (5.6%) | 1760 (14.5%) |
| Total | 5810 (47.8%) | 1405 (11.6%) | 1855 (15.3%) | 3085 (25.4%) | 12,155 (100%) |

Percentages for each cell are relative to the total of 12,155 immigrants.

[a]10 stateless refugees are excluded

foreign-born, there were 17 cases during 1997–2008 (age: 15–75 years, median 41) while 26 occurred in 2009–2020 (age: 19–78 years, median 32). Of these 43 cases among the foreign-born, respectively 35 (81%), 4 (9%), 4 (9%) and 0 occurred in nationals of countries with an incidence of ≥100, 50–99, 30–49 and <30 per 100,000 [15].

Among the 26 cases of TB diagnosed in foreign-born individuals after the opening of CDR, 7 had arrived prior to 2009, 8 arrived legally with a status other than refugee, one was an asylum seeker, 3 were refugees found to have TB upon screening at CDR, 3 were refugees never seen at CDR, 3 were seen at CDR with either a negative TST or a risk<5%, and one lived outside Quebec.

For Canada-born individuals, incidence decreased from 0.9 (1997–2008) to 0.3 per 100,000 person-years (2009–2020). In foreign-born individuals (excluding those from the USA or Western Europe), there was a modest drop in incidence, from 22.1 to 18.2 per 100,000.

More germane to the evaluation of our intervention, we measured incidence among immigrants from the 20 countries where our cohorts of refugees came from. Denominators included all categories since Statistics Canada does not distinguish refugees from others. In numerators, we eliminated 3 patients diagnosed with TB upon presentation at CDR, one student and one person living outside Quebec–who could not possibly have benefited from management of LTBI at CDR. Incidence decreased from 68.2 per 100,000 (1997–2008) to 26.3 per 100,000 (2009–2020). Assuming a stable incidence without the intervention, this suggests that 19 cases of TB were prevented so far.

Among national from 99 countries that did not provide refugees, and excluding persons born in the USA or Western Europe, the incidence remained stable: 11.7 per 100,000 person-years in 1997–2008 vs 9.7 per 100000 in 2009–2020.

## Activities at the refugee clinic

Between 2009 and 2020, 5131 refugees and asylum seekers were evaluated, aged between 6 weeks and 84 years. IRCC informs CDR of dates of arrival of incoming refugees, who are in time contacted to schedule a first appointment. Fig 1 summarizes the number of new patients seen annually and the number who settled in Estrie according to IRCC. More detailed records were available for 2016–2020 and during this period 3089/3251 (95.0%) refugees or asylum claimers who settled in Estrie were evaluated at CDR. In 2018–2019, additional human resources were provided to clear a backlog of patients. Seventy-three percent of refugees originated from countries with a TB incidence ≥100 per 100,000 (Afghanistan, Bhutan, and most of sub-Saharan Africa), 2% from countries with an incidence of 50–99 per 100,000 (Rwanda, Togo), 14% from countries with an incidence of 30–49 per 100,000 (Iraq, Colombia) and 11% from countries with an incidence <30 per 100,000 (Iran, Syria). Overall, roughly half of the refugees came from Afghanistan.

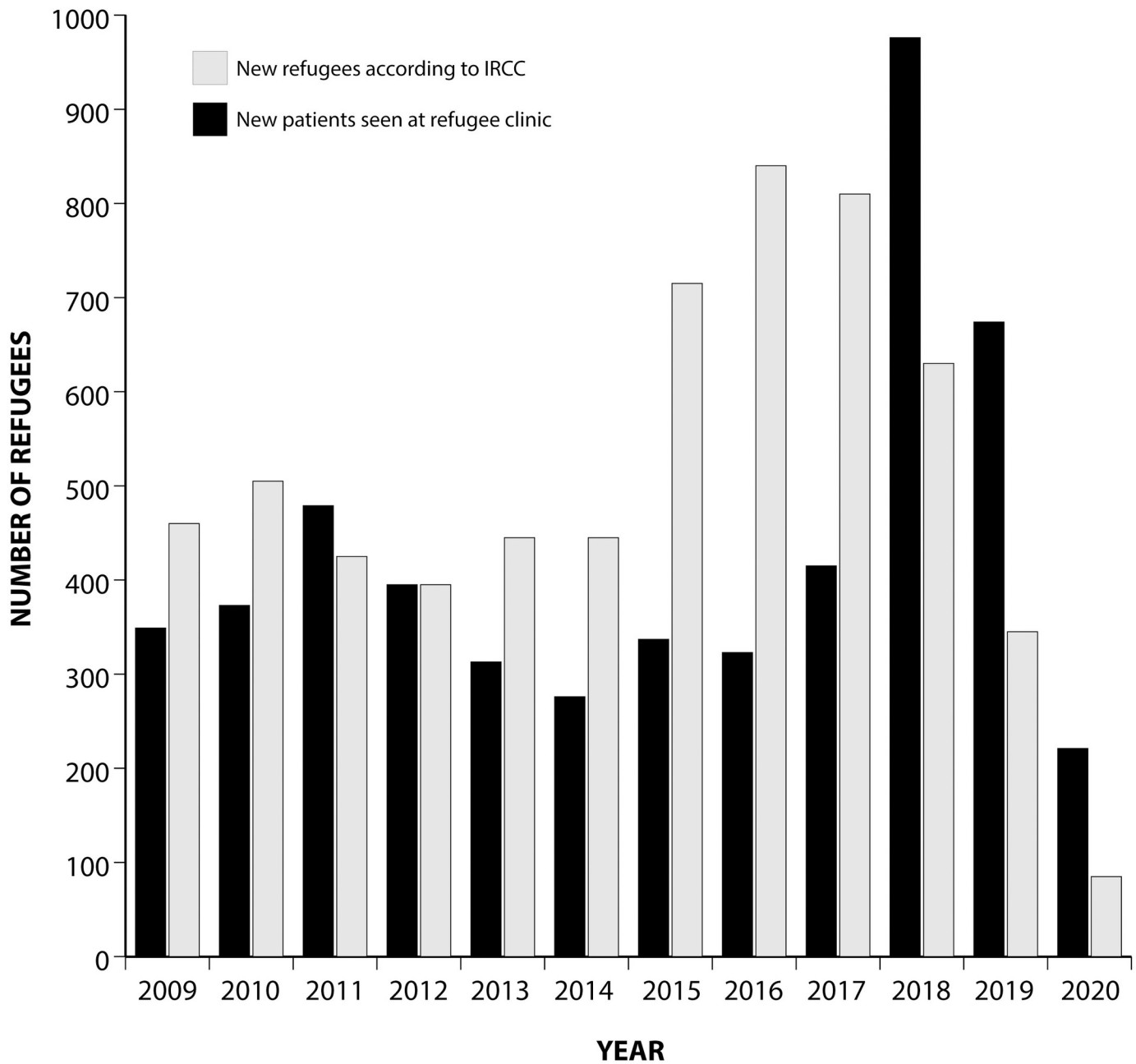

**Fig 1. Number of new patients seen at the refugee clinic and of refugees settling in Estrie, 2009–2020.**

## Management of latent tuberculosis infection

Table 3 summarizes data for the periods during which all CDR records were reviewed (1906 patients in 2010–2014, 1638 in 2018–2019). Losses to follow-up prior to initiation of LTBI treatment were uncommon and mostly due to patients moving out of the region. Prevalence of LTBI in adults (TST ≥10 mm without evidence of active TB) was 52% in 2010–14 and 38% in 2018–2019. When IGRA was more frequently used in 2018–2019 among the TST-positives, prevalence of LTBI in children was only 6% (43/687). Prevalence varied according to countries of origin, which changed over time. Among patients treated for LTBI, 85% completed the

**Table 3. Pathway of refugees evaluated for latent tuberculosis infection.**

| | 2010–2014 | 2018–2019 |
|---|---|---|
| **New patients** | 1906 | 1638 |
| **Excluded from analysis** | | |
| Past treatment for tuberculosis | 19 | 13 |
| Active tuberculosis | 0 | 1 |
| **TST done and read** | 1873/1887 (99%) | 1610/1624 (99%) |
| **TST ≥10 mm** | | |
| All | 723/1873 (38%) | 468/1610 (29%) |
| Adults | 560/1082 (52%) | 355/923 (38%) |
| Children | 163/791 (21%) | 113/687 (16%) |
| CXR done | 708/723 (98%) | 465/468 (99%) |
| **Referred for LTBI management based on a risk of reactivation of 5% or more** | 391 | 290 |
| **Evaluated** | 361/391 (92%) | 277/290 (96%) |
| **IGRA positive/IGRA total** | | |
| Adults | 8/19 (42%) | 31/79 (39%) |
| Children | 2/23 (9%) | 16/86 (19%) |
| **LTBI treatment** | | |
| Recommended | 295 | 146 |
| Deferred (pregnancy or breastfeeding) | 4 | 3 |
| Deferred because of other health issues | 12 | 3 |
| Deferred because patient about to move out | 7 | 6 |
| Not indicated, risk <5% | 11 | 1 |
| Not indicated, IGRA negative | 32 | 118 |
| **Treatment completion** | | |
| Rifampin throughout | 213/240 (89%) | 119/140 (85%) |
| Isoniazid throughout | 23/39 (59%) | 5/5 (100%) |
| Rifampin + isoniazid | 10/11 (91%) | |
| Rifampin, then isoniazid or vice-versa | 4/5 (80%) | 0/1 (0%) |
| Overall | 250/295 (85%) | 124/146 (85%) |
| **Mean lifetime risk of reactivation for patients with TST ≥10 mm[a]** | | |
| All adults | 7.1% | 8.0% |
| Adults non referred for LTBI | 4.3% | 3.8% |
| Adults treated for LTBI | 10.3% | 14.6% |
| Children treated for LTBI | 9.0% | 8.6% |
| Adults and children treated for LTBI | 10.0% | 13.5% |
| **For each case of TB prevented** | | |
| Number that needed to be screened | 83.8 | 107.6 |
| Number that needed to be treated | 13.1 | 9.7 |

[a] For adults, risk as calculated through TSTin3D [9]; for children, risk assumed to be 12% for those aged 0–5 years, then declining progressively to 6% for those aged 17

Abbreviations: TST: tuberculin skin testing; LTBI: latent tuberculosis infection; IGRA: interferon-gamma release assay; CXR: chest X-ray.

treatment both in 2010–2014 and in 2018–2019. As shown in Fig 2, compliance with the whole cascade amounted to 69.3%: 95.0% of refugees who settled in Estrie were seen at CDR * 99.2% of whom had a TST done and read * 98.5% of the latter had a CXR * 93.7% of those with a risk

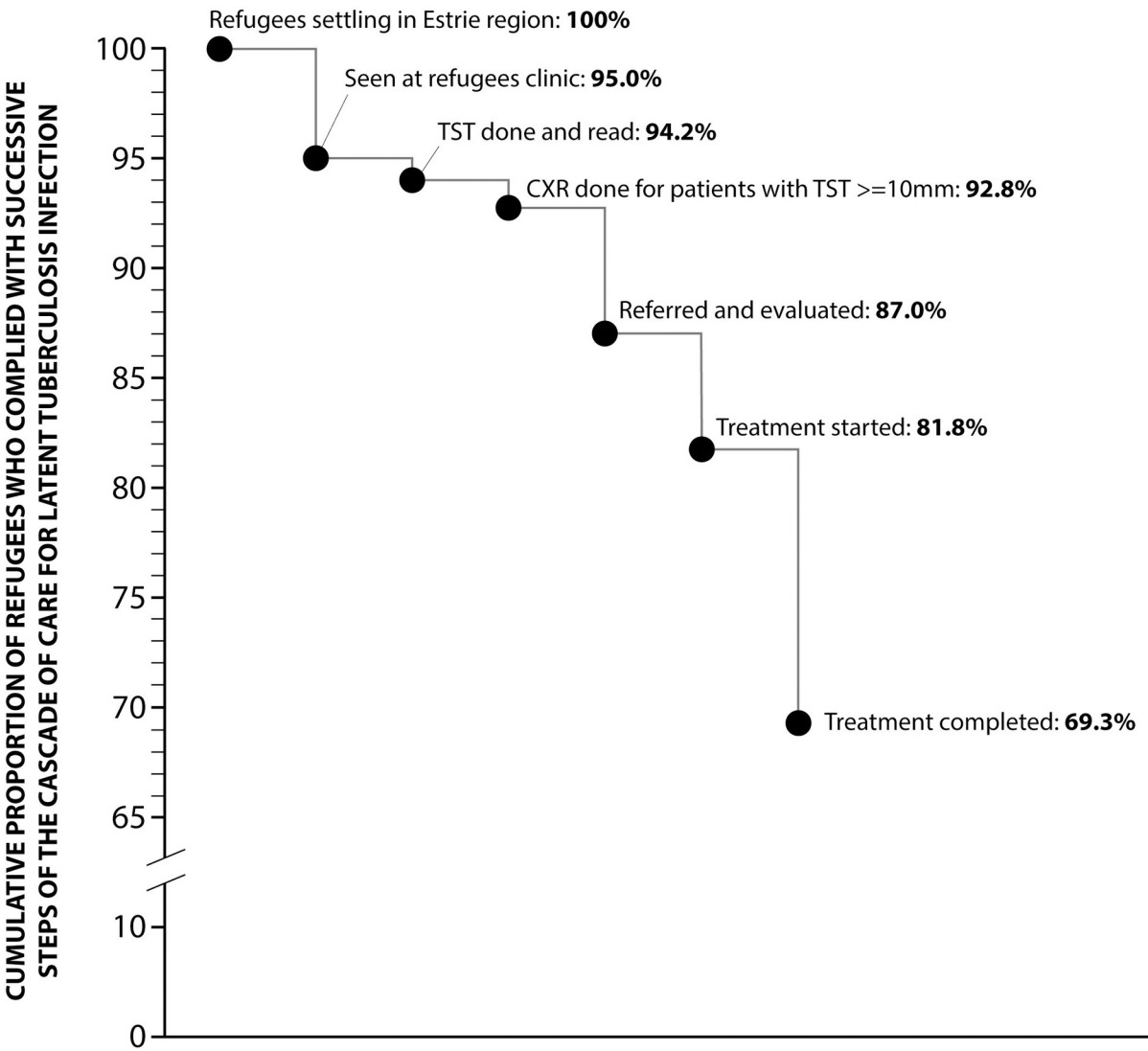

**Fig 2. Losses and drop-outs at each step of the cascade of care for the management of latent tuberculosis infection.** TST: tuberculin skin test. CXR: chest X-ray. The percentage at each level corresponds to the value from the preceding step multiplied by the estimate of compliance with the current step.

of reactivation of ≥5% were evaluated * 94.0% for whom treatment was started *84.8% whose treatment was completed.

## Benefit-cost analyses

Table 4 shows the costs incurred by the healthcare system for each case of TB in 2020 dollars (total: $37,583). Hospitalization costs represented three fourths of the total. Cost per person screened (Table 5) was $95 (only for TB screening, not the whole package). Cost for LTBI treatment was $590 per person.

We estimated how many refugees needed to be investigated for each case prevented, using data in Table 3 and combining both periods of observation. Overall, treatment of LTBI was initiated in 12.5% of refugees screened. Either 4 months of rifampin, 9 months of INH or 3 months of INH/rifampin are thought to have a similar efficacy ≈90% [5, 11, 12], which we

**Table 4. Costs for each case of active tuberculosis in 2020 Canadian dollars, based on 43 foreign-born patients seen in 2007–2020.**

| | | |
|---|---|---|
| Mycobacterial cultures and nucleic acid amplification tests | 3.0 positive cultures @ $128; 2.9 negative cultures @ $110; 0.4 blood culture @ $53; 1.0 PCR on culture @ $76; 0.4 PCR on clinical specimen @ $124; 1.0 PCR for rifampin resistance @ $126 | $976 |
| Per diem for hospitalization | Ward: 15.7 days (range 0–74, median 14) @ $1237; Critical care: 1.8 days @ $3948 | $26527 |
| Drugs | 2 HRZE, then 4HR | $807 |
| Medical honorarium in-hospital (pneumology + infectious diseases) | 2 consultations on day 1, 2 daily visits; bronchoscopy for 50% of cases | $1963 |
| Drug distribution by community pharmacies | $9 per drug * 16 distributions | $144 |
| X-rays and lab tests during follow-up | 3 Chest X-Rays; transaminase, complete blood count, creatinine x 8 | $300 |
| Medical honorarium, outpatient | 8 visits @ $43 | $344 |
| Public health investigation | | |
| Nurses | 18.2 contacts (range: 0–184), 2.5 hours/contact @ $59/hour | $2685 |
| Tuberculin | 18.2 contacts @ $15 per test | $273 |
| LTBI management among contacts | 18.2 contacts of whom 29 % are TST+ @ $560 | $2956 |
| Directly observed therapy by community pharmacist | 95 days on average @ $4.5/day | $428 |
| Translators | 6 hours @ $30/hour | $180 |
| Total | | **$37,583** |

Abbreviations: TST: tuberculin skin testing; LTBI: latent tuberculosis infection; PCR: polymerase chain reaction: H: isoniazid; R: rifampin: Z: pyrazinamide; E: ethambutol

**Table 5. Costs of identification and management of latent tuberculosis infection (in 2020 Canadian dollars).**

| | | |
|---|---|---|
| Screening at the refugee clinic, costs per person | | |
| Nursing time (TST, arrange CXR, risk calculation) | 0.5 hour/patient * $59 / hour | $30 |
| Medical honorarium | 10 minutes/patient @ $125 /hour | $21 |
| Tuberculin | $15 per patient | $15 |
| Chest X-Ray | 20 % have CXR @ $37 ($27 hospital cost + $10 for interpretation by radiologist) | $7 |
| CT scan | 2% have CT scan to rule out active TB $ ($62 hospital cost + $63 for interpretation by radiologist) and 30 minutes for further medical evaluation @ $125 /hour | $3 |
| Interferon-gamma release assay | 5.9% @ $72 | $4 |
| Translators | 30 minutes @ $30/hour | $15 |
| Total | | **$95** |
| Referrals and management of latent tuberculosis infection | | |
| Medical honorarium | 1 consultation, 1 follow-up visit | **$253** |
| Drugs | | |
| | 4 months of rifampin | $222 |
| | Distribution by pharmacists | $45 |
| Lab tests | alanine aminotransferase at 4 and 8 weeks | $40 |
| Translators | 1 hour @ $30/hour | $30 |
| Total | | **$590** |

Abbreviations: TST: tuberculin skin testing; CXR: chest X-ray.

used for refugees (85%) deemed of have completed treatment, while those with an incomplete course (15%) were considered to have 0% efficacy, yielding an overall efficacy of 76.5%. Among adults and children treated, the mean lifetime risk of reactivation was 11.0%.

Thus 95.1 (1/0.125/0.11/0.765) refugees needed to be screened and 11.9 (1/0.11/0.765) treated to prevent one case of TB. Adding the cost of screening ($95*95.1 = $9035) and LTBI treatment ($590*11.9 = $7021), the total was $16,056 per case prevented, for an undiscounted benefit-cost ratio of 2.34.

Using a 1.5% discounting rate [13], the value over a 30-year period of each case of TB averted from the perspective of LTBI cases managed in 2020 was $32,631, for a benefit-cost ratio of 2.03. With 3% discounting, the corresponding figures were $28,970 and 1.80.

The above data did not include the societal costs due to sick leaves, the cost of TB that would have developed in some contacts, nor the savings (shorter hospitalization, fewer infected contacts) made through a presumably earlier diagnosis of TB fr cases detected upon screening. A sensitivity analysis showed that the intervention was cost-beneficial if ≥4.4% of persons screened were treated with 11% average risk of reactivation, or if 12.5% were treated with 5.4% average risk of reactivation.

These estimates are based on the standard approach during the study period. For adults who received 9 months of INH, drug costs were lower ($183) but this was offset by longer follow-up and its related costs [16]. Using IGRA as a further screening assay among adults with a TST between 10–14 mm and in children was cost neutral. In 2018–19, when IGRA was used more frequently, the benefit-cost ratio remained unchanged: for each case prevented, the number needed to screen increased while the number needed to treat decreased.

## Discussion

In our region as elsewhere in the country, TB is fading away from Canadian-born non-indigenous individuals, as the older population infected in the remote past slowly disappears. Apart from the indigenous, TB will eventually be encountered only among the foreign-born fraction of the population. However, this will make it impossible for Canada to meet the objective of the 'pre-elimination phase', defined by the World Health Assembly as an incidence <1 per 100,000 by 2035 [17].

IRCC's current strategy for screening candidates is based on a CXR whose objective is to detect active TB (found in 0.05% of applicants) but not LTBI. This identifies some persons at high-risk for reactivation because they present fibronodular scarring but misses most cases of LTBI in individuals with a moderate risk (especially young adults with a normal CXR). Only 2% of individuals radiologically screened (aged ≥11 years) by IRCC are found to have "inactive TB" and referred for post-landing surveillance, a tiny fraction compared to the ≈50% of adults from endemic countries who have LTBI [4]. In Ontario, only 2.6% of cases of TB among immigrants were detected through the IRCC-mandated 2-year post-immigration surveillance [18]. Obviously, this approach cannot impact on the incidence of TB among migrants.

Since mid-2019, IRCC requires TST or IGRA for applicants who were close contacts of a case of TB within the past 5 years or had some other strong risk factors for reactivation (HIV, head and neck cancer, organ transplant, end-stage renal failure) [3]. However, applicants presenting at least one of these risk factors represent less than 5% of the future TB burden in immigrants [19]. Indeed, in our clinic, we have yet to see a single person referred by IRCC for LTBI.

Hoping to have a real impact on TB among immigrants, we thus developed a program to proactively diagnose and treat LTBI among newly arrived refugees. This intervention had a

favourable benefit-cost ratio. When applied to refugees specifically, for each dollar spent on LTBI management 2.03 dollars are saved in costs of active TB averted. Our estimate for the undiscounted cost per case of TB ($37,583), based on a small number of cases and largely driven by the cost of hospitalization, was very similar to one calculated from provincial/territorial expenditures: $38,935 in 2020 dollars (if calculated with our average number of contacts with or without LTBI) [20]. Our mean hospital stay (17.5 days) was higher than in a large series from Montreal (14.7 days) [21] as was the proportion of patients hospitalized (86% vs 54%), perhaps reflecting excessive caution by physicians who manage a small number of foreign-born patients and/or a better availability of hospital beds. Cost for LTBI treatment was identical to previous Canadian studies when corrected for inflation ($509-$572) while cost for screening was similar or cheaper than prior estimates ($67-$202) as it required little physician time [20, 22, 23].

Had our mean duration of hospitalization been the same as in Montreal, the cost per case of TB would have decreased to $33,905 or, after discounting, $29,438 with a benefit-cost ratio of 1.83. For this ratio to equal 1.0 and presuming that ICU stays are incompressible, it would be necessary that no stay on a ward occurs – which is of course impossible. Screening lower-risk immigrants (family reunifications, qualified workers, students) from high- or intermediate-incidence countries would increase cost per case prevented and the intervention may or may not remain cost-saving depending on the prevalence of LTBI, age at immigration, and other factors.

We presented herein results concerning the prevention of TB, but this activity was integrated within a package addressing multiple health issues. Evaluating the benefit-cost ratio of the whole intervention was beyond the scope of the current study, but other components are certainly cost-beneficial, for instance treating hepatitis C or hepatitis B or schistosomiasis before patients develop liver cirrhosis/fibrosis, treating HIV before a sexual contact is infected, or treating severe hypertension to avoid a stroke or renal failure. Other components did not generate savings but improved quality of life. Decision makers should consider the whole package.

We also showed that a program targeting recently arrived refugees is feasible and effective. Out of 5131 refugees screened only 3 subsequently developed TB. The sensitivity of TST for LTBI is ≈90%, like that of IGRA [24]. No patient treated for LTBI later developed TB, at least among those who remained in the region (it would have been tedious to link our database to the central registry for infectious diseases in Quebec, given confidentiality issues). Among migrants originating from the 20 countries that provided refugees, the incidence of TB decreased by 61%, which is a major achievement. Predictably, there was little change in TB incidence among nationals of countries that did not provide refugees–for whom no intervention was undertaken. However, among the latter, incidence was relatively low (9.7 per 100,000 in 2009–2020). Based on the numbers needed to prevent one case, the intervention may potentially prevent 54 cases of TB among the 5131 refugees and asylum seekers, over the lifetime of the persons screened. As mentioned previously, 19 cases of TB have possibly been already averted while most of the benefit is yet to come.

The TSTin3D calculator for risk of reactivation proved a valuable tool in identifying patients who would benefit the most from LTBI treatment [10]. TSTin3D is more attractive than another calculator that ignores risk factors such as radiological anomalies, diabetes, malnutrition, and smoking [25], but is perfectible. Individuals with a strongly positive TST seems at higher risk of reactivation, and TSTin3D could split the ≥15 mm into two strata. Among the TST-positives, TSTin3D provides no downward adjustment of risk if they have a negative IGRA, a shortcoming given that the latter experience a seven-fold lower incidence than those with a positive IGRA [26]. TSTin3D ends its tabulation of lifetime risk at age 80. This

underestimates the risk in the few refugees who arrive, as part of a larger family, at age 70 or 75, thus likely to survive beyond age 80 and at a substantial risk of reactivation given immune senescence. Current Canadian recommendations for screening are limited to refugees ≤50 years (≤65 years if they have some risk factor), which seems debatable given that the highest incidence occurs in individuals aged ≥55 years upon immigrating [4, 14, 27].

The calculator may overestimate the risk of reactivation in children, in immigrants from countries with an incidence <100 per 100,000 or where BCG is given beyond age 2 years, and those with a TST of 5–9 mm, all groups likely to have falsely positive TST post-BCG [28]. We did not offer treatment to anybody with a TST <10 mm, unless they had HIV infection, and most refugees came from countries with an incidence ≥100 per 100,000 where BCG is not given beyond the first year of life. It thus seems plausible that the overestimation of risk among our patients was modest.

Another limitation was that while our LTBI data used much larger populations than previous studies of refugees in Canada [29, 30], our measures around incident cases of TB were based on small numbers and lacked precision. Our assessment of patient compliance with LTBI treatment was imperfect. For practical reasons, our policy was not to see all adult patients at the very end of treatment; some who were compliant during the first two or three months may have later abandoned. Nevertheless, our 85% completion rate was consistent over time, and slightly superior to measures from other Canadian centres (73%-79%) [29, 30] and from a meta-analysis (73%) [6].

But because there were fewer losses before initiation of treatment, compliance with the whole cascade of care was 69% among our refugees, far better than the 14% found elsewhere in the same meta-analysis [6]. Key factors for this satisfactory compliance rate may include: i) close collaboration with community organizations that provide support to incoming refugees; ii) easy access to translators; iii) integration of the intervention within a package of medical care for other infectious and non-infectious diseases for the whole family; iv) its delivery shortly after arrival, when patients may be more compliant with medical recommendations; v) reasonable geographic access (most refugees lived within 8 kilometers of CDR); vi) preferential use of rifampin over INH, shortening treatment and enhancing completion rate; vii) risk-based selection of patients to be treated.

The dismal 14% compliance rate with the whole cascade in other settings explains why screening for LTBI in immigrants is generally considered ineffective [4–8]. Increasing five-fold this compliance rate obviously changes the equation, and we believe that our results could be replicated elsewhere. Throughout Quebec, a dozen or so centres are providing services targeting incoming refugees, and the critical point may be the rapid linking of new arrivals with the healthcare system through community organizations.

It would be possible to go further in preventing TB in migrants. Table 2 shows that a substantial number of nationals of countries with a TB incidence ≥50 per 100,000 landed as economic immigrants or through familial reunification, representing 37% of the arrivals. Apart from those with an abnormal CXR, these migrants are not linked with the health system upon arrival and there is no detection of LTBI. There are two potential solutions. First, all migrants from high-incidence countries (≥50 per 100,000) could be referred to the CDR or a similar provider upon landing in Canada where LTBI screening could be performed. In our case, this would represent 200–250 additional persons to be evaluated each year. Second, all adult migrants from high-incidence countries could undergo a TST as part of their pre-immigration medical examination, with post-landing referral of those with a risk of reactivation above a certain threshold (say, 5% or more). Merely adding a low-tech test to a system already deployed across the world would be less demanding for Canada-based healthcare facilities but would ignore health concerns other than LTBI.

Foreign-born persons long established in Canada represent a huge reservoir of LTBI (locally, ≈ 5500 foreign-born persons from high-incidence countries were living in Estrie in 2008). Annual incidence of TB decreases after 5 years post-landing [5, 14] but at this point those immigrants still have many years of life left ahead of them, and it can be calculated that half of incident cases occur more than 8 years post-landing. Healthcare providers following such patients for other medical conditions (specially those that enhance the risk of reactivation at least five-fold) [10] should seize this opportunity to test for LTBI. Obstacles in addressing this reservoir are many and it seems more realistic to focus on new arrivals.

## Conclusions

In conclusion, in a mid-size city of Canada, a program for treatment of LTBI among refugees proved comprehensive, efficacious, and cost-beneficial. Overall compliance with all steps of the process was 69%, far better than what has been reported in other jurisdictions. TB incidence in this subpopulation was reduced by 61%, very much in line with what could have been predicted given a treatment efficacy of 90% in those who complete therapy. Extending this intervention to other categories of migrants from high-incidence countries is needed in order to further reduce incidence of TB among the foreign-born.

Without a concerted effort, it will be impossible for Canada to meet its target for pre-elimination by 2035. Large-scale diagnosis and treatment of LTBI among high-risk new arrivals would impact more rapidly than waiting for the incidence of TB to drop in their home countries, an unlikely scenario for the foreseeable future in countries like Afghanistan, the Democratic Republic of Congo, and the Central African Republic, to name just a few. Due to Canada's geography, most of its immigrants are pre-selected and arrive within systematic relocation programs. In the United States, millions flow through its southern border, while in the European Union many countries are flooded with asylum seekers that display a considerable mobility as they look for the most welcoming jurisdiction. In this context, it should be easier for Canada to address the challenge of TB prevention among migrants. The most crucial necessary elements presently are political will, public health prioritization and adequate funding.

## Acknowledgments

We are grateful to Maude Gagnon, Geneviève De Bellefeuille and Lucie Kandu for their assistance in data collection, to François Milord and Irène Rose-Berthe Lamothe for their collaboration, to Geneviève Beaulieu and Cybèle Bergeron for their clinical contribution, and Philippe De Wals for helpful suggestions. We also thank Adrien Dubé, Lisa Landry, Céline Geoffroy, Sylvie Pothier, Carine Bouliane, Julie Laflamme and Julie Roy for their important contribution at the Clinique des Réfugiés.

## Author Contributions

**Conceptualization:** Jacques Pépin, France Desjardins, Louiselle LeBlanc.

**Data curation:** Jacques Pépin, France Desjardins, Alex Carignan, Michel Lambert, Isabelle Vaillancourt, Christiane Labrie, Dominique Mercier, Rachel Bourque, Louiselle LeBlanc.

**Formal analysis:** Jacques Pépin, France Desjardins, Alex Carignan, Louiselle LeBlanc.

**Methodology:** Jacques Pépin.

**Supervision:** Jacques Pépin.

**Writing – original draft:** Jacques Pépin, Louiselle LeBlanc.

**Writing – review & editing:** Jacques Pépin, France Desjardins, Alex Carignan, Michel Lambert, Isabelle Vaillancourt, Christiane Labrie, Dominique Mercier, Rachel Bourque, Louiselle LeBlanc.

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
