## [Decision Letter · Decision Letter 0]

21 Feb 2022

PONE-D-21-35333Impact and benefit-cost ratio of a program for the management of latent tuberculosis infection among refugees in a region of CanadaPLOS ONE

Dear Dr. Pépin,

Thank you for submitting your manuscript to PLOS ONE. After careful consideration, we feel that it has merit but does not fully meet PLOS ONE’s publication criteria as it currently stands. Therefore, we invite you to submit a revised version of the manuscript that addresses the points raised during the review process.

ACADEMIC EDITOR: Pls respond to the reviewers' comments.

We look forward to receiving your revised manuscript.

Kind regards,

Gang Qin, PhD, MD

Academic Editor

PLOS ONE

Journal Requirements:

Reviewers' comments:

Reviewer's Responses to Questions

**Comments to the Author**

1. Is the manuscript technically sound, and do the data support the conclusions?

Reviewer #1: Yes

Reviewer #2: Yes

2. Has the statistical analysis been performed appropriately and rigorously? 

Reviewer #1: No

Reviewer #2: Yes

3. Have the authors made all data underlying the findings in their manuscript fully available?

Reviewer #1: Yes

Reviewer #2: Yes

4. Is the manuscript presented in an intelligible fashion and written in standard English?

Reviewer #1: No

Reviewer #2: Yes

5. Review Comments to the Author

Reviewer #1: This is with reference to the manuscript entitled "Impact and benefit-cost ratio of a program for the management of latent tuberculosis infection among refugees in a region of Canada". The manuscript written by Jacques Pépin et al. is quiet interesting, well designed and planned for further execution. I appreciate to reply some minor comments reply from the authors which are mentioned below:

1. The authors are requested to revise the conclusion part in the Abstract portion in the above submitted Proposal.

2. Authors are requested to revise the methodology with detailed work plan describing the study population, duration of study, age of the participants, any standard proforma used for data collection, any specific exclusion criteria involved and statistical analysis used in the study as very limited information’s are available in the current submission.

3. Authors are requested to rewrite Line No. 83 and Line no. 218-220.

4. Authors are requested to add the Statistical analysis in the manuscript including the cost-utility analyses.

5. It would be better if the authors provide the flowchart of the study for better clarification as the above mentioned study is too vast and involved different duration and parameters.

6. The Author Contributions needs to be revised by the authors as these are not according to the Journal guidelines.

7. Lastly, the Authors are requested to please have a final look at the references mentioned as per the Journal guidelines.

Reviewer #2: This thesis has a good selection of topics, and has certain value in controlling the epidemic of tuberculosis. The data are detailed, the research plan is reasonable, the statistical methods are appropriate, the results are credible and the discussion is more comprehensive.

6. PLOS authors have the option to publish the peer review history of their article (what does this mean?). If published, this will include your full peer review and any attached files.

Reviewer #1: No

Reviewer #2: No

---

## [Author Response · Author response to Decision Letter 0]

14 Mar 2022

Dr Gang Qin, PhD, MD

Academic Editor

PLOS ONE

Dear Dr Qin

We are submitting a revised version of our manuscript PONE-D-21-35333. We are grateful for the comments made by your reviewers, and we feel that this version has been substantially improved compared to the original one.

You will find below ia point-by-point description of the changes made. We hoped that we interpreted correctly what Reviewer 1 had in mind.

We have also made changes to the format of tables so that they are in line with the PLoS One requirements.

We hope that you will be satisfied with this new version.

Best regards

Jacques Pépin, MD, MSc

Reviewers' comments:

Reviewer #1: This is with reference to the manuscript entitled "Impact and benefit-cost ratio of a program for the management of latent tuberculosis infection among refugees in a region of Canada". The manuscript written by Jacques Pépin et al. is quiet interesting, well designed and planned for further execution. I appreciate to reply some minor comments reply from the authors which are mentioned below:

1. The authors are requested to revise the conclusion part in the Abstract portion in the above submitted Proposal.

We have revised the abstract. The conclusion is now shorter. Indeed, some elements that used to be in the conclusion belonged to the methods. Thanks for this suggestion.

2. Authors are requested to revise the methodology with detailed work plan describing the study population, duration of study, age of the participants, any standard proforma used for data collection, any specific exclusion criteria involved and statistical analysis used in the study as very limited information’s are available in the current submission.

Some additions have been made at various places in the manuscript. The age range is given, we specify that there were no exclusion criteria. We included refugees seen in 2009-2020, and cases of TB that were diagnosed between 1997 and 2020, but this was already spelled out in the text. Data were collected with a simple Excel spreadsheet. We now specify in the abstract that this was a purely descriptive study, i.e. without statistical analyses aiming to prove or disprove a hypothesis. 

3. Authors are requested to rewrite Line No. 83 and Line no. 218-220.

We have removed the sentence around line 83 and shortened the caption for Figure 1 (lines 218-220).

4. Authors are requested to add the Statistical analysis in the manuscript including the cost-utility analyses.

We have modified the benefit cost section and moved it to a different place in the manuscript. We have also added some methodological details for this section. We hope that this is now OK

5. It would be better if the authors provide the flowchart of the study for better clarification as the above mentioned study is too vast and involved different duration and parameters.

We have added a flowchart as Figure 2, which summarizes compliance with various steps of the ‘cascade of care’. It complements what is elsewhere in the manuscript and makes it easier for readers to follow the flow of patients

6. The Author Contributions needs to be revised by the authors as these are not according to the Journal guidelines.

These have been corrected.

7. Lastly, the Authors are requested to please have a final look at the references mentioned as per the Journal guidelines.

We have modified the references and are confident that this is now in line with PLoS requirements. We have added the DOI and PMID (apart from two papers for which there are no DOI as per PubMed). Please note that in the marked-up version, we accepted all modifications for the reference list. Otherwise, there were too many corrections, and it became very difficult to sort out if it was OK. We did not add or remove references compared to the original version. 

Reviewer #2: This thesis has a good selection of topics, and has certain value in controlling the epidemic of tuberculosis. The data are detailed, the research plan is reasonable, the statistical methods are appropriate, the results are credible and the discussion is more comprehensive.

Thanks !

---

## [Editor Report · Decision Letter 1]

18 Apr 2022

Impact and benefit-cost ratio of a program for the management of latent tuberculosis infection among refugees in a region of Canada

PONE-D-21-35333R1

Dear Dr. Pépin,

We’re pleased to inform you that your manuscript has been judged scientifically suitable for publication and will be formally accepted for publication once it meets all outstanding technical requirements.

Kind regards,

Gang Qin, PhD, MD

Academic Editor

PLOS ONE

---

## [Editor Report · Acceptance letter]

20 Apr 2022

PONE-D-21-35333R1 

Impact and benefit-cost ratio of a program for the management of latent tuberculosis infection among refugees in a region of Canada 

Dear Dr. Pépin:

I'm pleased to inform you that your manuscript has been deemed suitable for publication in PLOS ONE. Congratulations! Your manuscript is now with our production department. 

Kind regards, 

on behalf of

Dr. Gang Qin 

Academic Editor

PLOS ONE